# Association between Vitamin D and Short-Term Functional Outcomes in Acute Ischemic Stroke

**DOI:** 10.3390/nu15234957

**Published:** 2023-11-29

**Authors:** Min-Su Kim, Jin San Lee, Sung Joon Chung, Yunsoo Soh

**Affiliations:** 1Department of Physical Medicine & Rehabilitation, Kyung Hee University Hospital, College of Medicine, Kyung Hee University, Seoul 02447, Republic of Korea; 25496@khmc.or.kr (M.-S.K.); csjk@khnmc.or.kr (S.J.C.); 2Department of Neurology, Kyung Hee University Hospital, College of Medicine, Kyung Hee University, Seoul 02447, Republic of Korea; xpist@khu.ac.kr

**Keywords:** vitamin D, acute ischemic stroke, functional outcomes

## Abstract

Vitamin D (Vit D) affects musculoskeletal performance and central nervous system neuroprotection. We aimed to investigate the association between serum Vit D levels and short-term functional outcomes in patients with acute ischemic stroke. This study involved patients with acute ischemic stroke confirmed on brain MRI. The National Institutes of Health Stroke Scale (NIHSS) was used to assess initial stroke severity upon admission. We evaluated the functional outcomes using the Berg Balance Scale (BBS), Manual Function Test (MFT), Korean Mini-Mental State Examination (K-MMSE), Korean version of the modified Barthel Index (K-MBI) within three weeks from the onset of stroke, and modified Rankin Scale (mRS) score at discharge. Overall, 192 patients were finally included and divided into three groups: Vit D sufficient (*n* = 28), insufficient (*n* = 49), and deficient (*n* = 115). Multivariate analysis showed that the Vit D deficient group presented with a higher risk of initially severe stroke (*p* = 0.025) and poor functional outcomes on the BBS (*p* = 0.048), MFT (*p* = 0.017), K-MMSE (*p* = 0.001), K-MBI (*p* = 0.003), and mRS (*p* = 0.032) compared to the Vit D sufficient group. Vit D deficiency may be associated with severe initial stroke and poor short-term post-stroke functional outcomes.

## 1. Introduction

Stroke, the second leading cause of death, is a major cause of severe disability, and its economic burden continues to grow worldwide [1]. Strokes could be classified into hemorrhagic or ischemic stroke depending on the etiology, of which ischemic strokes are the most common type. Furthermore, their incidence has increased over the past few years [2]. Although proper management, including thrombolysis and rehabilitation, can improve the outcomes of ischemic stroke, many survivors experience long-term disabilities. Several factors are associated with the prognosis of ischemic stroke, including the etiology, lesion size and location, age, sex, obesity, and other medical comorbidities [3]. Recently, researchers have investigated various biological markers that can be directly obtained from blood samples for the prognosis of ischemic stroke [4,5].

Vitamin D (Vit D), also known as calciferol, is a fat-soluble compound that acts as a steroid hormone to regulate calcium homeostasis and bone metabolism [6]. In humans, Vit D is mainly synthesized when the skin is exposed to sunlight and brings its biological activity by binding to the Vit D receptor [7]. Several studies have demonstrated that Vit D levels affect bone health and musculoskeletal performance [8]. Recently, the role of Vit D as a neuroprotective agent in the central nervous system and skeletal muscle function has gained increasing attention [9]. Previous studies have reported that Vit D deficiency is more prevalent in patients with ischemic compared to the general population [10]. Vit D deficiency is also associated with an increased initial stroke severity and mortality [11]. The prognosis after ischemic stroke is also related to various diseases accompanying the disease. In the case of depression, which can affect the prognosis of stroke patients, the risk of post-stroke depression was higher in patients with Vit D deficiency [12]. Infectious complications such as post-stroke pneumonia have been associated with Vit D deficiency [13]. Vit D levels affect stroke recurrence and functional outcomes [14].

Previous studies have provided limited information regarding the relationship between Vit D and the physical and cognitive performance of patients with stroke [15,16]. This study aimed to investigate the association between serum Vit D levels and short-term functional outcomes in acute ischemic stroke patients by using several different outcome measurements.

## 2. Materials and Methods

### 2.1. Study Population and Design

In this retrospective study, we included patients diagnosed with acute ischemic stroke on the brain MRI in the Kyung Hee University Hospital, Seoul, South Korea, from June 2022 to August 2023 [17]. Several factors that could affect the initial function of each stroke patient needed to be excluded [18]. The exclusion criteria were as follows: (1) patients who were admitted 7 days after stroke onset; (2) prior history of a cerebrovascular disease; (3) malignant tumor; (4) severe renal or hepatic disease; (5) neurodegenerative disorder such as Parkinsonism or dementia; (6) endocrine disorder; (7) taking Vit D, calcium supplementation, or steroid use; (8) impaired physical condition with a modified Rankin Score (mRS) ≥ 3 prior to stroke; (9) missing laboratory data; (10) missing functional outcome data. A flow diagram of the patient selection process is shown in Figure 1.

Baseline characteristics, including demographic and clinical data, were collected from the medical records. Demographic characteristics included age, sex, and BMI. To reflect the seasonal variation in serum Vit D levels due to changes in the amount of sunlight exposure, the season at which serum Vit D was collected was also considered: spring (from March to May), summer (from June to August), autumn (from September to November), and winter (from December to February) [19]. Medical histories, including diabetes mellitus, hypertension, dyslipidemia, smoking status, and alcohol consumption prior to stroke were recorded. Blood sampling for laboratory data was performed in all patients within 24 h of admission.

Total serum 25-hydroxy Vit D (25-OHD) level was measured as an indicator of each patient’s Vit D status, since it can reflect the amount of Vit D production from both the skin and dietary sources [20]. The biological half-life of 25-OHD is known to be 2–3 weeks, making its role as a useful long-term biomarker [21]. We analyzed a serum total of 25- hydroxy vit D (a sum of 25(OH) Vit D2 and D3) by chemiluminescence immunoassay on the Beckman Coulter Unicel DXI 800 (Beckman Coulter, Brea, CA, USA). Although the gold standard method for analyzing the serum Vit D is liquid chromatography–tandem mass spectrometry (LC–MS/MS) [22], Unicel DXI 800 immunoassay is moderately concordant with LC–MS/MS (concordance correlation coefficient = 0.916) and the total coefficient of variation (CV) of this assay were 8.3% at 31.7 ng/mL and 7% at 66.8 ng/mL [23]. Although the specific serum 25-OHD level representing Vit D deficiency varies according to each society’s guideline, our study adopted to use the definition from the clinical practice guideline of Endocrine Society, which has been widely used as a criteria for the treatment and prevention of Vit D deficiency [24]. The cutoff values from this guideline was based on several studies regarding the relationship between the Vit D and the serum level of parathyroid hormone and the amount of intestinal calcium absorption [21]. Vit D sufficiency was defined as a serum 25-OHD level > 30 ng/mL, insufficiency as 30 ng/mL ≥ 25-OHD ≥ 20 ng/mL, and deficiency as 25-OHD < 20 ng/mL [25]. Other laboratory data included complete blood counts (white blood cell count, hemoglobin concentration, and platelet count) and serum concentrations of high-sensitivity *C*-reactive protein (hsCRP), hemoglobin A1c, total cholesterol, triglycerides, high-density lipoproteins, and low-density lipoproteins. Reperfusion treatment and the length of hospital stay for each patient were also recorded.

Patients performed bedside or gym-based rehabilitation physical therapy daily, weekday, for at least 30 min to 2 h. Neurodevelopmental treatment (NDT), including mat exercise, gait training, muscle strengthening, stretching, balance training, and activity of daily living training, including transfer grooming, dressing, and hand skill training, were performed depending on the patient’s functional condition.

This study was approved by the Institutional Review Board of Kyung Hee University Hospital (KHUH 2023-10-018). Due to the retrospective design of this study, the requirement for informed consent from each patient was waived.

### 2.2. Outcome Measurements

To determine the initial stroke severity, the National Institutes of Health Stroke Scale (NIHSS) score was measured on admission. The NIHSS is the most widely used clinical rating scale for neurological deficits in stroke patients, reflecting several domains, including motor, sensory, and language declines [26]. Mild stroke was defined as an NIHSS score < 5, whereas moderate-to-severe stroke was defined as an NIHSS score ≥ 5 [27]. Patients underwent physical and cognitive function tests within three weeks of stroke onset. The Berg Balance Scale (BBS), which is a valid instrument that reflects both the static and dynamic balance of patients with stroke, was used to evaluate the balance ability. A BBS score ≤ 20, which generally indicates the requirement for a wheelchair for mobility and represents a higher fall risk, was considered a poor outcome [28]. To evaluate upper extremity function, the Manual Function Test (MFT) was performed on the hemiplegic side. MFT is a valid instrument for evaluating the upper limb motor function of patients with hemiplegic stroke; an MFT score ≤ 19, which reflects poor dexterity, was defined as a poor outcome [29]. Cognitive function was assessed using the Korean Mini-Mental State Examination (K-MMSE), a validated version of the MMSE widely used to screen for cognitive decline in Korea. A K-MMSE score ≤ 23, the level indicating cognitive impairment, was defined as a poor outcome [30]. Overall activities of daily living performance was assessed using the Korean version of the modified Barthel Index (K-MBI). The K-MBI demonstrates the functional independence of individual patients in several domains of daily activities in Korea. A K-MBI score ≤ 50, which indicates severe functional dependence, was considered a poor outcome [31]. At discharge, the modified Rankin scale (mRS) score, which reflects general disability and functional dependency, was evaluated. An mRS score ≥ 3, representing moderate-to-severe disability, was defined as a poor outcome [32]. Experienced clinicians, physical therapists, and occupational therapists evaluated the NIHSS, BBS, MFT, K-MMSE, K-MBI, and mRS scores of each patient.

### 2.3. Statistical Analysis

The Kolmogorov–Smirnov test was used to assess the normal distribution of variables. Variables with a normal distribution were assessed using analysis of variance (ANOVA). Variables with a non-normal distribution were assessed using Kruskal–Wallis test. The chi-square and linear-by-linear association tests were used for the comparison of categorical variables. The Spearman’s correlation test was used to determine the correlation between two continuous variables. Univariate and multivariate logistic analyses were performed to evaluate the association between serum Vit D levels and clinical outcomes. Adjusting for potential confounding factors, including age, sex, BMI, season, hypertension, diabetes mellitus, dyslipidemia, coronary artery disease, smoking, alcohol consumption, intravenous thrombolysis, endovascular therapy, time from stroke onset to hospital admission, white blood cell count, hemoglobin concentration, platelet count, and serum concentrations of hsCRP, hemoglobin A1c, total cholesterol, triglycerides, high-density lipoproteins, and low-density lipoproteins, was performed in the multivariate analysis. In order to determine the optimal serum Vit D cut-off levels for increased stroke severity and poor functional outcomes, we performed receiver operating characteristic (ROC) curve analysis. SPSS Statistics version 25 (SPSS Inc., Chicago, IL, USA) was used for statistical analyses. Two-sided *p*-values < 0.05 were considered statistically significant.

## 3. Results

Our study included 192 patients (Figure 1) who were divided into 3 groups based on serum Vit D levels at admission: Vit D sufficient group (25-OHD > 30 ng/mL, *n* = 28), Vit D insufficient group (30 ng/mL ≥ 25-OHD ≥ 20 ng/mL, *n* = 49), and Vit D deficient group (25-OHD < 20 ng/mL, *n* = 115). Table 1 presents the baseline characteristics of the patients included in this study according to Vit D levels. The median age of all of the patients was 74 years (range, 64–83 years), and 78 (40.6%) patients were female. The three groups had significant seasonal differences; the Vit D sufficient and insufficient groups were more prevalent in the summer, whereas the Vit D deficient group was more prevalent in the winter. The median serum 25-OHD level was 17.1 ng/mL (range, 12.2–24.2 ng/mL) for all patients, 40.1 ng/mL (range, 35.7–43.0 ng/mL) for the Vit D sufficient group, 23.4 ng/mL (range, 21.5–27.3 ng/mL) for the Vit D insufficient group, and 13.1 ng/mL (range, 9.3–16.6 ng/mL) for the Vit D deficient group. Other characteristics, including demographic factors, medical history, laboratory findings, time from stroke onset to hospital admission, length of hospital stay, and reperfusion treatments, showed no significant difference between each group.

The odds ratios (OR) for the initial stroke severity and functional outcomes among the three groups are presented in Table 2. In the univariable analysis, the Vit D deficient group had higher risks of initially severe stroke, as measured by the NIHSS (OR, 4.07; 95% CI, 1.45–11.45; *p* = 0.008) and poor outcomes on the BBS (OR, 4.00; 95% CI, 1.30–12.29; *p* = 0.015), MFT (OR, 5.36; 95% CI, 1.53–18.79; *p* = 0.009), K-MMSE (OR, 5.57; 95% CI, 1.98–15.68; *p* = 0.001), K-MBI (OR, 5.38; 95% CI, 1.91–15.14; *p* = 0.001), and mRS (OR, 3.60; 95% CI, 1.36–9.54; *p* = 0.010) compared with the Vit D sufficient group. The results were similar in the multivariate analysis when confounding factors were adjusted. In multivariate analysis, the Vit D deficient group had higher risks of initially severe stroke (OR, 4.98; 95% CI, 1.22–30.33; *p* = 0.025) and poor outcomes on the BBS (OR, 3.88; 95% CI, 1.01–14.51; *p* = 0.048), MFT (OR, 5.69; 95% CI, 1.36–23.83; *p* = 0.017), K-MMSE (OR, 9.80; 95% CI, 2.40–39.93; *p* = 0.001), K-MBI (OR, 7.13; 95% CI, 1.98–25.59; *p* = 0.003), and mRS (OR, 3.58; 95% CI, 1.12–11.49; *p* = 0.032) compared with the Vit D sufficient group. However, the Vit D insufficient group showed no significant differences in initial stroke severity and functional outcomes compared with the Vit D sufficient group.

Patients were divided into two groups, and a comparison between the Vit D deficient group (25-OHD < 20 ng/mL) and Vit D non-deficient group (25-OHD ≥ 20 ng/mL) is presented in Table 3. Except for the BBS, the results showed a significant difference between the Vit D deficient and non-deficient groups in multivariate analysis.

Table 4 summarizes the correlations among serum Vit D levels, initial stroke severity, and functional outcomes. Significant positive correlations (i.e., clinically poor outcomes) were identified between serum Vit D levels and BBS (ρ = 0.200, *p* = 0.006), MFT (ρ = 0.334, *p* < 0.001), K-MMSE (ρ = 0.297, *p* < 0.001), and K-MBI (ρ = 0.311, *p* < 0.001) scores. Significant negative correlations (i.e., clinically poor outcomes) were identified between serum Vit D levels and NIHSS (ρ = −0.402, *p* < 0.001) and mRS (ρ = −0.274, *p* < 0.001) scores.

Receiver operating characteristic (ROC) curve analysis was used to determine the optimal serum Vit D cut-off values for stroke severity and functional outcomes (Figure 2). The serum Vit D cut-offs were 17.22 ng/mL for initially severe stroke, 17.05 ng/mL for poor BBS scores, 16.06 ng/mL for poor MFT scores, 18.27 ng/mL for poor K-MMSE scores, and 17.14 ng/mL for poor K-MBI and mRS scores. Based on these cutoffs, additional logistic regression analysis was performed to determine the relationship between serum Vit D levels and stroke severity and functional outcomes. In multivariate analysis, lower serum Vit D levels were associated with increased risks of stroke severity (OR 6.02; 95% CI, 2.34–15.50; *p* < 0.001) and poor functional outcomes based on the BBS (OR 2.47; 95% CI, 1.12–5.44; *p* = 0.025), MFT (OR 2.33; 95% CI, 1.07–5.10; *p* = 0.033), K-MMSE (OR 4.83; 95% CI, 1.92–12.18; *p* = 0.001), K-MBI (OR 3.17; 95% CI, 1.41–7.17; *p* = 0.005), and mRS (OR 2.71; 95% CI, 1.24–5.94; *p* = 0.013) scores (Table 5).

## 4. Discussion

The present study analyzed the relationship between serum Vit D levels and short-term functional outcomes of patients with acute ischemic stroke using multiple indicators. In multivariate logistic regression analyses, Vit D deficiency was associated with an initially severe stroke, as measured by NIHSS scores, and poor functional outcomes based on the BBS, MFT, K-MMSE, K-MBI, and mRS scores. The optimal serum Vit D cutoff values based on ROC curve analysis of increased initially severe stroke and poor functional outcomes ranged from 16.06–18.27 ng/mL. In addition, multivariate regression analysis based on these cut-offs revealed significant associations with lower serum Vit D levels. These results are consistent with those of previous studies that demonstrated an association between Vit D deficiency and poor short-term post-stroke functional outcomes [11,14].

These results can be explained by the role of Vit D in the pathogenesis of ischemic stroke. During the acute phase of ischemic brain injury, damage-associated molecular patterns (DAMPs) are released from the insulted brain cells, which trigger the local immune response [33]. DAMPs promotes the recruitment and activation of several inflammatory cells, including neutrophils, lymphocytes, and macrophages [34]. Various types of pro-inflammatory cytokines including interleukin-6 (IL-6), interleukin-1β (IL-1β), and tumor necrosis factor-α (TNF-α) are produced that further accelerate the infiltration of leukocytes to the site of damaged brain tissue, which then produce reactive oxygen species and inflammatory mediators, resulting in the breakdown of the blood-brain barrier [35]. Vit D attenuates ischemic brain injury by promoting anti-inflammatory responses through several mechanisms. Calcitriol, which is an active form of Vit D, binds to the Vit D receptor and execute its biological function through the genomic transcription [36]. Calcitriol promotes the function of regulatory T cells by regulating the expression of transcription factors, which produce anti-inflammatory cytokines such as interleukin-10 (IL-10) and suppress the release of pro-inflammatory cytokines [37]. Furthermore, Vit D alleviates ischemic damage by upregulating antioxidant processes and interrupting the production of reactive oxygen species, thereby lowering the oxidative stress in the brain tissue [38]. Vit D also participates in activating detoxification pathways and producing several neurotrophic factors, including insulin-like growth factor-I (IGF-I) and nerve growth factor (NGF). Altogether, these processes help to regenerate neurons in damaged brain tissue and promote recovery after ischemic stroke [39].

In addition, Vit D affects the cerebrovascular system through several mechanisms. Vit D might be involved in the autoregulation of cerebral blood flow, possibly by influencing the myogenic tone and the vascular resistance of cerebral blood vessels [40]. Vit D also modulates the vascular remodeling process through vascular mediators such as vascular endothelial growth factor (VEGF) and matrix metalloproteinases (MMPs) [41]. Vit D influences the homeostasis of the extracellular matrix and vascular wall stiffness, and Vit D deficiency contributes to vascular calcification [42]. The integrity and permeability of the blood-brain barrier can be affected by Vit D via the expression of tight junction proteins [43]. Taken together, Vit D deficiency could be related to the dysregulation of cerebral blood flow and vascular endothelial cell dysfunction, making the cerebral tissue more vulnerable to ischemic insults. This could help to explain the reason why Vit D deficiency was associated with increased initial severity of acute ischemic stroke in our study.

Vit D is also important for maintaining the functional integrity of the musculoskeletal system. Vit D is important for the regulation of bone and calcium metabolism, and Vit D deficiency is associated with low bone mineral density [6]. Vit D is thought to be involved in myogenesis and skeletal muscle regeneration, and Vit D deficiency can lead to muscular atrophy [44]. In particular, Vit D affects type II muscle fibers, also known as fast-twitch muscle fibers, which are important for the generation of rapid force and maintenance of appropriate posture [45]. Vit D deficiency could be related to poor balance and an increased risk of falls in the elderly [46]. Serum Vit D levels affect musculoskeletal performance in athletic populations, including the vertical jump height, handgrip strength, and maximal running capacity [8]. Therefore, declines in musculoskeletal function may cause functional declines during the early stages of ischemic stroke.

The effect of Vit D on neuroprotection of the central nervous system and musculoskeletal function could help explain our finding that Vit D deficiency is associated with poor functional outcomes in patients with acute ischemic stroke. While prior studies demonstrated the relationship between serum Vit D levels and post-stroke outcomes, functional assessment tools were limited to a few indicators, such as MBI or mRS scores. This study evaluated the functional outcomes of patients with ischemic stroke in several clinical domains using different indicators. To the best of our knowledge, our study is the first to demonstrate the relationship between serum Vit D levels and the physical performance of stroke patients by measuring MFT and BBS scores. Other studies have attempted to determine the optimal vit D cut-off values for the clinical outcomes of patients with ischemic stroke using ROC analysis. One study reported that a serum Vit D level ≤ 17 ng/mL, measured by enzyme-linked immunosorbent assay (ELISA), predicts poor post-stroke outcomes based on mRS scores three months after the stroke, with a sensitivity of 81.4% and specificity of 42.7% (AUC = 0.654, *p* < 0.001) [47]. It is intriguing to see that this Vit D cutoff was similar with the cutoff from our study results, which was 17.14 ng/mL for poor mRS scores. Another study reported that the optimal serum Vit D cutoff value, measured by chemiluminescence immunoassay on the E601 modular, for predicting stroke recurrence is 11.7 ng/mL, with a sensitivity of 82.3% and specificity of 64.5% (AUC = 0.82, *p* < 0.001) [48]. The serum Vit D cutoff values vary depending on the different post-stroke outcomes and study settings, including the methods used for measuring the serum Vit D levels. However, these results are similar in that low serum Vit D levels are associated with unfavorable clinical outcomes. Further research is required to determine the optimal cutoff level of serum Vit D for the poor post-stroke functional outcomes with standardized methods using ROC analysis.

The present study did not address the effects of Vit D supplementation in patients with ischemic stroke. A recently published narrative review revealed that Vit D supplementation may have positive effects on the functional outcomes of patients with ischemic stroke throughout rehabilitation [10]. Some studies showed that the beneficial effect of Vit D supplementation for post-stroke outcomes was statistically significant, while others did not [49,50,51,52,53,54]. Our study differs from the Vit D supplementation studies as we excluded patients with Vit D supplementation. However, we examined the initial severity and short-term outcomes related to the neuroprotective effect of Vit D on the degree of damage at the time of ischemic brain injury. Our study results might help to provide the background explanation for future studies regarding Vit D supplementation on stroke patients.

Our study had several limitations. First, owing to the retrospective nature of the study conducted in a single hospital, selection bias may have occurred. The sample size of this study was limited, and only 28.5% of the total stroke patients satisfied both of our study’s inclusion and exclusion criteria. A larger sample size could provide better reliability of the study results regarding the relationship between serum Vit D levels and post-stroke outcomes. Second, we only considered the Vit D status of patients at the time of admission. Temporal changes in the serum level of Vit D were not reflected in our study, such as the level before the onset of stroke and at discharge from the hospital. If these data were available, the relationship between Vit D status and post-stroke functional outcomes could be better explained. Third, the size, location, and etiology of acute ischemic stroke were not considered in this study. These factors may have acted as potential complications in the study results. Additional research, such as propensity score matching, may be necessary, depending on the acute ischemic stroke size, location, and etiology.

## 5. Conclusions

The serum Vit D level is a useful biomarker that can be directly obtained from blood samples. The present study demonstrated that Vit D deficiency might be associated with poor short-term functional outcomes in patients with acute ischemic stroke. Further studies regarding the beneficial effect of Vit D supplementation on post-stroke functional outcomes are required.

## Figures and Tables

**Figure 1 nutrients-15-04957-f001:**
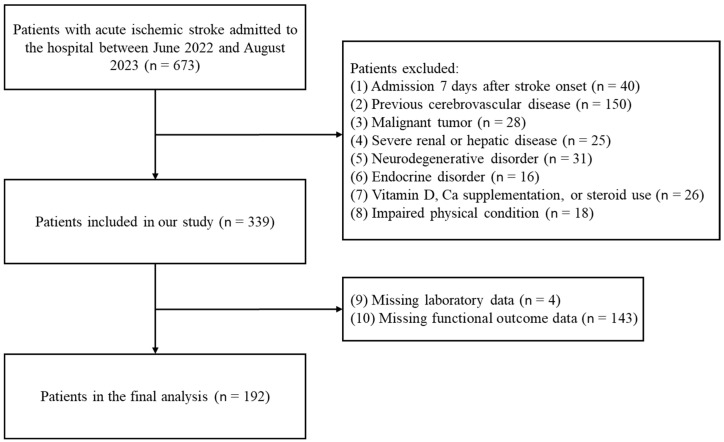
Flow diagram for patient selection.

**Figure 2 nutrients-15-04957-f002:**
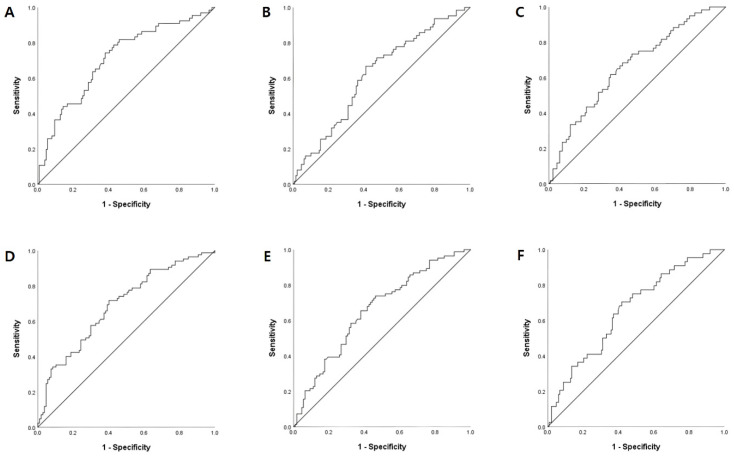
Receiver operating characteristic curve analyses for the associations of serum vitamin D levels with initial stroke severity and functional outcomes. (**A**) NIHSS (AUC: 0.71; cut-off: 17.22 ng/mL; sensitivity: 0.74; specificity: 0.38; *p* < 0.001), (**B**) BBS (AUC: 0.62; cut-off: 17.05 ng/mL; sensitivity: 0.67; specificity: 0.41; *p* = 0.008), (**C**) MFT (AUC: 0.67; cut-off: 16.06 ng/mL; sensitivity: 0.62; specificity: 0.35; *p* < 0.001), (**D**) K-MMSE (AUC: 0.69; cut-off: 18.27 ng/mL; sensitivity: 0.72; specificity: 0.40; *p* < 0.001), (**E**) K-MBI (AUC: 0.66; cut-off: 17.14 ng/mL; sensitivity: 0.66; specificity: 0.38; *p* < 0.001), (**F**) mRS (AUC: 0.65; cut-off: 17.14 ng/mL; sensitivity: 0.66; specificity: 0.39; *p* = 0.001). Abbreviations: NIHSS, National Institutes of Health Stroke Scale; BBS, Berg Balance Scale; MFT, Manual Function Test; K-MMSE, Korean Mini-Mental State Examination; K-MBI, Korean version of the modified Barthel Index; mRS, modified Rankin Scale; AUC, area under the curve.

**Table 1 nutrients-15-04957-t001:** Baseline characteristics of patients with acute ischemic stroke according to the level of vitamin D.

	Total	Vit D Sufficient Group (25-OHD > 30 ng/mL)	Vit D insufficient Group (30 ng/mL ≥ 25-OHD ≥ 20 ng/mL)	Vit D Deficient Group (25-OHD < 20 ng/mL)	
	(*n* = 192)	(*n* = 28)	(*n* = 49)	(*n* = 115)	*p*-Value
Demographics					
Age (years)	74.0 (64.0–83.0)	73.5 (66.8–83.3)	74.0 (66.0–82.0)	73.0 (63.0–83.0)	0.957
Female sex, *n* (%)	78 (40.6)	16 (57.1)	17 (34.7)	45 (39.1)	0.136
BMI (kg/m^2^)	24.0 ± 3.3	24.3 ± 2.9	23.9 ± 3.3	24.0 ± 3.5	0.876
Season					0.017 *
Spring, *n* (%)	33 (17.2)	6 (21.4)	8 (16.3)	19 (16.5)	
Summer, *n* (%)	72 (37.5)	12 (42.9)	25 (51.0)	35 (30.4)	
Autumn, *n* (%)	39 (20.3)	7 (25.0)	9 (18.4)	23 (20.0)	
Winter, *n* (%)	48 (25.0)	3 (10.7)	7 (12.3)	38 (33.0)	
Medical history					
Hypertension, *n* (%)	114 (59.4)	20 (71.4)	29 (59.2)	65 (56.5)	0.354
Diabetes mellitus, *n* (%)	69 (35.9)	13 (46.4)	18 (36.7)	38 (33.0)	0.413
Dyslipidemia, *n* (%)	71 (37.0)	16 (57.1)	15 (30.6)	40 (34.8)	0.050
Coronary artery disease, *n* (%)	24 (12.5)	4 (14.3)	6 (12.2)	14 (12.2)	0.953
Smoking, *n* (%)	65 (33.9)	7 (25.0)	17 (34.7)	41 (35.7)	0.559
Alcohol consumption, *n* (%)	67 (34.9)	8 (28.6)	19 (38.8)	40 (34.8)	0.664
Intravenous thrombolysis, *n* (%)	20 (10.4)	3 (10.7)	6 (12.2)	11 (9.6)	0.875
Endovascular therapy, *n* (%)	34 (17.7)	3 (10.7)	7 (14.3)	24 (20.9)	0.346
Onset to admission time (days)	2 (1–3)	2 (1–2)	2 (1–3)	2 (1–3)	0.979
Length of hospital stay (days)	24 (19–35)	23 (20–35)	25 (20–34)	24 (19–37)	0.954
Laboratory findings					
25-OHD (ng/mL)	17.1 (12.2–24.2)	40.1 (35.7–43.0)	23.4 (21.5–27.3)	13.1 (9.3–16.6)	<0.001 ***
Hemoglobin (g/dL)	13.7 ± 1.8	13.2 ± 2.0	13.9 ± 1.5	13.7 ± 1.9	0.253
Platelet (10^9^/L)	224 (174–256)	221 (173–253)	215 (171–250)	228 (183–261)	0.581
WBC (10^9^/L)	6.9 (5.6–8.7)	6.6 (5.0–7.7)	6.9 (6.1–8.2)	7.3 (5.6–9.0)	0.156
hsCRP (mg/L)	0.2 (0.1–0.5)	0.1 (0.0–0.3)	0.1 (0.1–0.3)	0.2 (0.1–0.6)	0.194
HbA1c (%)	5.8 (5.4–6.5)	6.0 (5.5–7.2)	5.8 (5.4–6.6)	5.8 (5.4–6.4)	0.359
TC (mg/dL)	178 (150–203)	163 (125–206)	176 (154–196)	183 (152–204)	0.403
TG (mg/dL)	115 (84–175)	115 (87–152)	106 (80–145)	120 (86–190)	0.429
HDL (mg/dL)	45 (38–52)	45 (37–52)	49 (40–54)	45 (37–52)	0.224
LDL (mg/dL)	104 (82–123)	93 (69–123)	103 (89–119)	106 (82–126)	0.508

* *p* < 0.05, *** *p* < 0.001. Note: Continuous variables with normal distributions are presented as means ± standard deviations. Continuous variables with non-normal distributions are presented as medians (interquartile ranges). Categorical variables are presented as numbers and percentages. Abbreviations: BMI, body mass index; 25-OHD, 25-hydroxy vitamin D; WBC, white blood cell; hsCRP, highly sensitive *C*-reactive protein; HbA1c, hemoglobin A1c; TC, total cholesterol; TG, triglycerides; HDL, high-density lipoprotein; LDL, low-density lipoprotein; Vit D, vitamin D.

**Table 2 nutrients-15-04957-t002:** Univariate and multivariate logistic analyses of initial stroke severity and functional outcomes according to sufficient, insufficient, and deficient serum vitamin D levels.

	Vit D Sufficient Group (25-OHD > 30)	Vit D Insufficient Group (30 ≥ 25-OHD ≥ 20)	Vit D Deficient Group (25-OHD < 20)
	(*n* = 28)	(*n* = 49)	(*n* = 115)
Moderate-to-severe stroke severity (NIHSS ≥ 5)			
Unadjusted OR (95% CI)	ref	0.77 (0.22–2.69), *p* = 0.609	4.07 (1.45–11.45), *p* = 0.008 **
Adjusted OR (95% CI)	ref	0.66 (0.13–3.28), *p* = 0.609	4.98 (1.22–20.33), *p* = 0.025 *
Poor BBS score (BBS ≤ 20)			
Unadjusted OR (95% CI)	ref	2.17 (0.63–7.44), *p* = 0.219	4.00 (1.30–12.29), *p* = 0.015 *
Adjusted OR (95% CI)	ref	2.33 (0.57–9.62), *p* = 0.242	3.83 (1.01–4.51), *p* = 0.048 *
Poor MFT score (MFT ≤ 19)			
Unadjusted OR (95% CI)	ref	2.70 (0.69–10.56), *p* = 0.153	5.36 (1.53–18.79), *p* = 0.009 **
Adjusted OR (95% CI)	ref	3.09 (0.69–13.89), *p* = 0.141	5.69 (1.36–23.83), *p* = 0.017 *
Poor K-MMSE score (MMSE ≤ 23)			
Unadjusted OR (95% CI)	ref	2.44 (0.79–7.58), *p* = 0.122	5.57 (1.98–15.68), *p* = 0.001 **
Adjusted OR (95% CI)	ref	3.93 (0.90–17.14), *p* = 0.068	9.80 (2.40–39.93), *p* = 0.001 **
Poor K-MBI score (MBI ≤ 50)			
Unadjusted OR (95% CI)	ref	2.44 (0.79–7.58), *p* = 0.122	5.38 (1.91–15.14), *p* = 0.001 **
Adjusted OR (95% CI)	ref	3.11 (0.81–11.93), *p* = 0.099	7.13 (1.98–25.59), *p* = 0.003 **
Poor mRS score (mRS ≥ 3)			
Unadjusted OR (95% CI)	ref	1.79 (0.60–5.23), *p* = 0.297	3.60 (1.36–9.54), *p* = 0.010 *
Adjusted OR (95% CI)	ref	2.07 (0.58–7.35), *p* = 0.261	3.58 (1.12–11.49), *p* = 0.032 *

* *p* < 0.05, ** *p* < 0.01. Adjusted for age, sex, BMI, season, hypertension, diabetes mellitus, dyslipidemia, coronary artery disease, smoking, alcohol consumption, intravenous thrombolysis, endovascular therapy, onset-to-admission time, and hemoglobin, platelet, WBC, hsCRP, HbA1c, TC, TG, HDL, and LDL levels. Abbreviations: NIHSS, National Institutes of Health Stroke Scale; BBS, Berg Balance Scale; MFT, Manual Function Test; K-MMSE, Korean Mini-Mental State Examination; K-MBI, Korean version of the modified Barthel Index; mRS, modified Rankin Scale; OR, odds ratio; CI, confidence interval; Vit D, vitamin D.

**Table 3 nutrients-15-04957-t003:** Univariate and multivariate logistic analyses of initial stroke severity and functional outcomes according to non-deficient and deficient serum vitamin D levels.

	Vit D Non-Deficient Group (25-OHD ≥ 20 ng/mL)	Vit D Deficient Group (25-OHD < 20 ng/mL)
	(*n* = 77)	(*n* = 115)
Moderate-to-severe stroke severity (NIHSS ≥ 5)		
Unadjusted OR (95% CI)	ref	4.80 (2.34–9.82), *p* < 0.001 ***
Adjusted OR (95% CI)	ref	6.52 (2.40–17.73), *p* < 0.001 ***
Poor BBS score (BBS ≤ 20)		
Unadjusted OR (95% CI)	ref	2.35 (1.22–4.53), *p* = 0.010 *
Adjusted OR (95% CI)	ref	2.11 (0.94–4.73), *p* = 0.069
Poor MFT score (MFT ≤ 19)		
Unadjusted OR (95% CI)	ref	2.66 (1.35–5.23), *p* = 0.005 **
Adjusted OR (95% CI)	ref	2.55 (1.10–5.89), *p* = 0.029 *
Poor K-MMSE score (MMSE ≤ 23)		
Unadjusted OR (95% CI)	ref	3.03 (1.64–5.61), *p* < 0.001 ***
Adjusted OR (95% CI)	ref	3.94 (1.58–9.83), *p* = 0.003 **
Poor K-MBI score (MBI ≤ 50)		
Unadjusted OR (95% CI)	ref	2.93 (1.58–5.41), *p* = 0.001 **
Adjusted OR (95% CI)	ref	3.31 (1.45–7.56), *p* = 0.004 **
Poor mRS score (mRS ≥ 3)		
Unadjusted OR (95% CI)	ref	2.46 (1.33–4.54), *p* = 0.004 **
Adjusted OR (95% CI)	ref	2.24 (1.02–4.92), *p* = 0.045 *

* *p* < 0.05, ** *p* < 0.01, *** *p* < 0.001. Adjusted for age, sex, BMI, season, hypertension, diabetes mellitus, dyslipidemia, coronary artery disease, smoking, alcohol consumption, intravenous thrombolysis, endovascular therapy, onset-to-admission time, and hemoglobin, platelet, WBC, hsCRP, HbA1c, TC, TG, HDL, and LDL levels. Abbreviations: NIHSS, National Institutes of Health Stroke Scale; BBS, Berg Balance Scale; MFT, Manual Function Test; K-MMSE, Korean Mini-Mental State Examination; K-MBI, Korean version of the modified Barthel Index; mRS, modified Rankin Scale; OR, odds ratio; CI, confidence interval; Vit D, vitamin D.

**Table 4 nutrients-15-04957-t004:** Correlation of serum vitamin D levels with initial stroke severity and functional outcomes.

Variables	Spearman’s Correlation Coefficient (ρ)	*p*-Value
Vit D-NIHSS	−0.402	<0.001 ***
Vit D-BBS	0.200	0.006 **
Vit D-MFT	0.334	<0.001 ***
Vit D-K-MMSE	0.297	<0.001 ***
Vit D-K-MBI	0.311	<0.001 ***
Vit D-mRS	−0.274	<0.001 ***

** *p* < 0.01, *** *p* < 0.001. Abbreviations: Vit D, vitamin D; NIHSS, National Institutes of Health Stroke Scale; BBS, Berg Balance Scale; MFT, Manual Function Test; K-MMSE, Korean Mini-Mental State Examination; K-MBI, Korean version of the modified Barthel Index; mRS, modified Rankin Scale.

**Table 5 nutrients-15-04957-t005:** Logistic regression based on optimal serum vitamin D cut-off levels for initial stroke severity and functional outcomes.

Variables	AUC (95% CI)	Cut-off Value (ng/mL)	Sensitivity	Specificity	Unadjusted OR (95% CI)	Adjusted OR (95% CI)
NIHSS	0.71 (0.63–0.79), *p* < 0.001 ***	17.22	0.74	0.38	4.68 (2.42–9.05), *p* < 0.001 ***	6.02 (2.34–15.50), *p* < 0.001 ***
BBS	0.62 (0.54–0.70), *p* = 0.008 **	17.05	0.67	0.41	2.87 (1.53–5.39), *p* = 0.001 **	2.47 (1.12–5.44), *p* = 0.025 *
MFT	0.67 (0.58–0.74), *p* < 0.001 ***	16.06	0.62	0.35	3.01 (1.60–5.66), *p* = 0.001 **	2.33 (1.07–5.10), *p* = 0.033 *
K-MMSE	0.69 (0.62–0.76), *p* < 0.001 ***	18.27	0.72	0.40	3.78 (2.06–6.96), *p* < 0.001 ***	4.83 (1.92–12.18), *p* = 0.001 **
K-MBI	0.66 (0.58–0.73), *p* < 0.001 ***	17.14	0.66	0.38	3.10 (1.71–5.62), *p* < 0.001 ***	3.17 (1.41–7.17), *p* = 0.005 **
mRS	0.65 (0.57–0.73), *p* = 0.001 **	17.14	0.66	0.39	3.02 (1.66–5.50), *p* < 0.001 ***	2.71 (1.24–5.94), *p* = 0.013 *

* *p* < 0.05, ** *p* < 0.01, *** *p* < 0.001. Adjusted for age, sex, BMI, season, hypertension, diabetes mellitus, dyslipidemia, coronary artery disease, smoking, alcohol consumption, intravenous thrombolysis, endovascular therapy, onset-to-admission time, and hemoglobin, platelet, WBC, hsCRP, HbA1c, TC, TG, HDL, and LDL levels. Abbreviations: NIHSS, National Institutes of Health Stroke Scale; BBS, Berg Balance Scale; MFT, Manual Function Test; K-MMSE, Korean Mini-Mental State Examination; K-MBI, Korean version of the modified Barthel Index; mRS, modified Rankin Scale; AUC, area under the curve; OR, odds ratio; CI, confidence interval.

## Data Availability

Data are contained within the article.

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
