# Peer review of "Association between Vitamin D and Short-Term Functional Outcomes in Acute Ischemic Stroke"

_nutrients, 2023, doi:10.3390/nu15234957_

Round 1
Reviewer 1 Report
Comments and Suggestions for Authors
This is a well planned and described retrospective study about the role of Vit D levels in patients with stroke. Vit D deficiency was associated with severe initial stroke and poor functional outcomes in patienta with acute ischemic stroke. Patients selection and description is good as well as the number of subjects (192). The relationship between serum Vit D levels and the physical performance of patients were demonstrated and optimal Vit D values were suggested. Limits were correctly declared.
So minor points for improving the ms should be: a more clear description of the reason why sufficient, insufficient or deficient Vit D values where individuated as follows: "Vit D sufficiency was defined as a serum 25-OHD level >30 ng/mL, insufficiency 76 as 30 ng/mL ≥25-OHD ≥20 ng/mL, and deficiency as 25-OHD <20 ng/mL"
The discussion of the molecular explanation why VitD should be associated with a more severe intial stroke should be improved.
The authors could be discuss the point: "how long should be the period of life with deficient VitD levels to increase the risk of worse response to stroke?
Reviewer 2 Report
Comments and Suggestions for Authors
The study by Kim. Et al. entitled “Vitamin D as a Predictor for Short-term functional outcomes in acute ischemic stroke”, focused on the Vitamin D association with initial stroke severity and short-term functional evaluation.
This retrospective study is intriguing and significant because it is time to look into the relationship between biological parameters of nutritional health and post-stroke recovery.
There are, however, a few major criticisms that authors should address in order to improve the manuscript, especially for data presentation.
First of all, the title is not correct: I suggest something like “Association between Vitamin D and functional outcomes in acute ischemic stroke”.
The term “predictor” is too strong, and the first important question is: what kind of prediction is vitamin D made?
In fact, a two-week functional status assessment is not related to rehabilitation treatment, and in the outcome measurements, the authors did not mention any treatment. They evaluated patients at admission, and with physical and cognitive function tests, and they evaluated mRS at discharge (but they did not describe the timing of discharge in the study population and design section).
Introduction:
The literature and background are coherent but it seems a list of sentences, without any connection or conjunction between sentences. it would be necessary to make the text more harmonious.
The authors refer to “previous study, without citing the bibliography (line 50).
Material and Methods:
The declaration of Helsinki is in the wrong section (at the end of outcome measurements) and should be moved to the “study population and design” paragraph, and there is no reference to the code of clinicaltrial.gov. Did the authors declare this or register on?
The inclusion and exclusion criteria appear to be described in a confused and fragmented manner.
There is not a clear explanation of the clinical management of the patients; when they did the ematochemical blood collection and analysis; if they underwent some type of rehabilitation; and when they were discharged. The authors should explain it clearly.
The seasonality related to vitamin D should be better clarified: do they refer to the seasonality of the blood sample? should be specified.
It should also be clearly stated which method was used to test for Vitamin D (Serum total 25-OHD (the sum of 25-OHD3 and 25-OHD2); did they employ chemiluminescence immunoassay, or ELISA immunoassay o did they employ the newly liquid chromatography-mass spectrometry (LC-MS) or tandem mass spectrometry? Some assays, even when based on a similar methodology, are less accurate and precise, making comparison of data across assays or laboratories difficult. The gold standard analytical method is LC-MS/MS and that should always be the method of choice for national surveys. However, even the LC-MS/MS method must meet the assay standardization criteria of the VDSP. Serum total 25(OH)D measurements can be 'prospectively' standardized or 'retrospectively' standardized, using methods developed by VDSP; see ( Sempos et al., 2017) Authors should report in the manuscript the CV of assay employed.
Statistical analysis: did the authors adjust data also for the season at admission /hematochemical analysis period?
RESULTS: In Table 1 some variables are reported as (n) but the number refers both to n and (%), please correct this; 25-OH Vit D lacks measurements unit (ng/mL), please add it in Table 1.
Table 4, please include in the table the variable Vit D, for correlation (not only in the explanation of the table).
The ROC curve did not report sensibility and specificity corresponding to the Vit D cut-off.
Discussion: the entire discussion is a list of references related to vitamin D, authors should instead comment on references and discuss them in relation to their important finding (ex. compare the ROC data reported in the results in relation to the cited literature ROC data).
Some sentences are too strong and too definitive, such as in lines 312-313.
The point is that is important to know the association between vitamin D and functional outcomes related to early stages of acute ischemic stroke. But first of all, the Authors should describe better the biomarker vitamin D. It is known that the plasma 25(OH)D concentration reflects medium to long-term vitamin D availability from both dietary and endogenous sources, thus making it the best biomarker of vitamin D exposure and status. But they should report it in the manuscript. Authors think that measuring 25(OH)D it is possible to understand the short-term functional outcome. So, a supplementation could ameliorate recovery, or the 25(OH)D is too near to functional outcome and could not be modified?
This is explained in the limitation of the study but it is not so clear… they deliberately excluded patients with supplementation, and this choice was correct, so this is not a limit. Or they can include it and evaluate it in relation to the short-term, but this choice needs extra control (supplementation, how long patients assumed supplementation, etc).
Conclusion: Vit D, 25(OH)vit D is not so simple and reliable biomarker. It depends on the kind of assay. Probably it cannot substitute the short-term functional assessment, but could be employed together and the clinician could prescribe the intake of supplementation.
Round 2
Reviewer 2 Report
Comments and Suggestions for Authors
I appreciate the work of Authors, and this revised manuscript is now ready for publication